# Sexually dimorphic control of gene expression in sensory neurons regulates decision-making behavior in *C. elegans*

Zoë A Hilbert, Dennis H Kim*

Department of Biology, Massachusetts Institute of Technology, Cambridge, United States

**Abstract** Animal behavior is directed by the integration of sensory information from internal states and the environment. Neuroendocrine regulation of diverse behaviors of *Caenorhabditis elegans* is under the control of the DAF-7/TGF-$\beta$ ligand that is secreted from sensory neurons. Here, we show that *C. elegans* males exhibit an altered, male-specific expression pattern of *daf-7* in the ASJ sensory neuron pair with the onset of reproductive maturity, which functions to promote male-specific mate-searching behavior. Molecular genetic analysis of the switch-like regulation of *daf-7* expression in the ASJ neuron pair reveals a hierarchy of regulation among multiple inputs—sex, age, nutritional status, and microbial environment—which function in the modulation of behavior. Our results suggest that regulation of gene expression in sensory neurons can function in the integration of a wide array of sensory information and facilitate decision-making behaviors in *C. elegans*.

## Introduction

In order to effectively adapt to an environment, organisms need to be able to analyze both their internal state and external surroundings to make behavioral decisions that will maximize their chances of survival. It has been hypothesized that decision making in animals is controlled primarily at the level of command interneurons or decision centers in the central nervous system, which receive input from an array of sensory neurons and integrate the information conveyed to inform behavioral decisions (*Kristan, 2008*). However, recent work in a number of animal species has demonstrated that modulation of gene expression and neuronal activity by both environmental cues and internal state can occur at the level of sensory neurons, suggesting that aspects of behavioral decision making may also occur in the peripheral nervous system (*Dey et al., 2015*; *Farhadian et al., 2012*; *Ryan et al., 2014*).

Two pervasive internally derived modifiers of decision making in animals are the nutritional state of the animal and biological sex. Feeding status and satiety levels have long been known to have dramatic effects on the behaviors and decisions of species across the animal kingdom (*Pool and Scott, 2014*; *Sengupta, 2013*). In many cases, the nutritional state of the animal will serve as a source of regulation directly on feeding and other food-related behaviors. For instance, in the medicinal leech, periods of food deprivation can lead to a heightened response to appetitive stimuli in its environment, a response that has been shown to be carried out via a correlation between serotonin production and time of last meal (*Gaudry and Kristan, 2012*; *Groome et al., 1993*; *Lent et al., 1991*). Starvation or nutritional deprivation can also have less direct effects on feeding behaviors as in *Drosophila*, where periods of starvation have been demonstrated to both enhance odor sensitivity and attraction as well as abrogate avoidance responses to normally aversive stimuli (*Bräcker et al., 2013*; *Inagaki et al., 2014*; *Ko et al., 2015*; *Root et al., 2011*). Similarly, the

*For correspondence: dhkim@mit.edu

Competing interests: The authors declare that no competing interests exist.

**eLife digest** For almost all species of animal, males and females will often behave differently in similar situations. Little is known about how these sex-specific differences are generated or, for example, how different the nervous system of a male is to that of a female. Moreover, it is also poorly understood how these underlying differences based on the biological sex of an animal are integrated with and influenced by its experiences and environment.

The roundworm *Caenorhabditis elegans* has two sexes, hermaphrodites and males. The male worms behave differently to the hermaphrodites in a number of situations. This means that these animals offer the opportunity to explore and understand sex-specific differences in behavior. It is also possible to analyze the underlying factors that contribute to behavior in *C. elegans*, because it has a relatively simple and well-defined nervous system.

Now, Hilbert and Kim show that a signal that influences how *C. elegans* explores in response to chemicals in its environment is expressed differently in male and hermaphrodite worms. The signal in question is molecule called DAF-7, which is released by several sensory neurons—nerve cells that are used for detecting cues from the environment. The sensory neurons that release DAF-7 are found in both sexes of *C. elegans* but the specific way that the male worms express this signal encourages them to search for mates. Hermaphrodites, on the other hand, do not need to search for mates because they can fertilize their own eggs.

Hilbert and Kim showed that the biological sex in combination with multiple other inputs – including the animal's past diet and age – regulate how the DAF-7 signal is expressed in *C. elegans*. These inputs all converge onto a single pair of sensory neurons, which integrate the inputs and enable the worm to assess its current and past experiences and alter its behavior accordingly.

Moving forward the next challenge is to understand how information about both external environment and internal states, such as hunger, are communicated to and integrated by these sensory neurons. Decoding the signals behind this process may illuminate how biological sex and internal states influence behavior in other species of animals.

biological sex of an animal is a potent modifier of decision-making. Not only does the sex of the animal lead to differences in mating behaviors and pheromone attraction, but studies in several systems have suggested that biological sex can also alter olfactory and gustatory preferences, perhaps to guide decision-making in the context of mating and reproduction (*Dey et al., 2015*; *Kimchi et al., 2007*; *Kurtovic et al., 2007*; *Lebreton et al., 2015*; *Lee and Portman, 2007*; *Meunier et al., 2000*; *Nakagawa et al., 2005*; *Ribeiro and Dickson, 2010*; *Ryan et al., 2014*; *Stowers and Logan, 2010*; *Stowers et al., 2002*; *Vargas et al., 2010*).

The roundworm, *Caenorhabditis elegans*, and its well-defined nervous system provide a simple experimental system well suited to investigate how internal states and environmental cues can be integrated to modulate animal behavior. *C. elegans* exist as one of two sexes: hermaphrodite or male. Hermaphrodites produce a limited amount of sperm during larval development making them capable of self-fertilization in addition to cross-fertilization. In contrast, males are only cross-fertile and must find mating partners for reproduction. As such, the male displays a large behavioral repertoire primarily aimed at facilitating mate finding and successful mating encounters. These behaviors include complex mating rituals (*Liu and Sternberg, 1995*), differential sensitivity to hermaphrodite pheromone (*Srinivasan et al., 2008*), and a mate-searching behavior in adult male animals that is driven by the prioritization of mating over feeding (*Lipton et al., 2004*). Mate-searching behavior of *C. elegans* has thus emerged as an interesting case study of decision making in a simple animal, which is subject to regulation by multiple features of internal state—sex, age, and nutritional status—as well as by external sensory experiences such as the presence of food or hermaphrodites in the environment (*Barrios et al., 2008*; *Lipton et al., 2004*; *Ryan et al., 2014*).

Here, we focus on the DAF-7/TGF-$\beta$ signaling pathway, which plays a pivotal role in the regulation of diverse aspects of *C. elegans* physiology and behavior including the dauer developmental decision, reproductive egg laying, feeding and foraging, fat storage, satiety quiescence, longevity, aggregation, aerotaxis and avoidance of pathogenic bacteria (*Chang et al., 2006*; *Gallagher et al.,*

*2013*; *Greer et al., 2008*; *Meisel et al., 2014*; *Milward et al., 2011*; *Ren et al., 1996*; *Shaw et al., 2007*; *White and Jorgensen, 2012*; *You et al., 2008*). The expression pattern of *daf-7* was originally identified as being restricted to a single bilaterally symmetric pair of neurons in the head of the worm, the ASI chemosensory neurons, but it is also expressed in the ADE and OLQ neurons under normal laboratory growth conditions (*Meisel et al., 2014*; *Ren et al., 1996*; *Schackwitz et al., 1996*). We recently defined an altered expression pattern for *daf-7* in hermaphrodites in response to exposure to bacterial metabolites produced by the pathogenic bacteria *Pseudomonas aeruginosa* (*Meisel et al., 2014*). Within minutes of being exposed to *P. aeruginosa*, *daf-7* expression is induced in an additional pair of chemosensory neurons, the ASJ neurons, which in turn promotes behavioral avoidance of the pathogenic bacterial lawn (*Meisel et al., 2014*). Here, we show that in male *C. elegans*, *daf-7* expression in the ASJ neuron pair is not only induced in response to environmental cues such as *Pseudomonas aeruginosa*, but can be switched 'on' or 'off' depending on multiple internal states such as sex, age, and experience. The differential effects of internal states and environmental experience on the DAF-7 transcriptional switch in the two ASJ sensory neurons reveal a hierarchy among different inputs that can function to modulate behavior.

## Results

### Male-specific expression of *daf-7*/TGF-β in the ASJ neurons

While expression of the DAF-7/TGF-$\beta$ ligand was long thought to be confined to the ASI chemosensory neurons (*Ren et al., 1996*; *Schackwitz et al., 1996*), we recently reported that a change in the microbial environment of the worm—namely, the introduction of the pathogen *P. aeruginosa*—can alter the neuronal expression pattern of *daf-7*, inducing expression in the ASJ neuron pair of hermaphrodite animals (*Meisel et al., 2014*). In the current study, we observed that adult males exhibit *daf-7* expression in both the ASI and the ASJ neuron pairs in the absence of *P. aeruginosa* (*Figure 1A and B*). This expression pattern difference in adult *C. elegans,* can be observed both through the use of a *pdaf-7::gfp* transcriptional reporter (*Figure 1A and B*) as well as by fluorescence in situ hybridization (FISH) that directly probes for endogenous *daf-7* mRNA (*Figure 1—figure supplement 1A–C*).

Recent studies have suggested that, in addition to the generation of sex-specific neurons late in larval development, both male-specific and sex-shared neurons exhibit changes in gene expression as well as synaptic remodeling as the male undergoes the sexual maturation process (*Oren-Suissa et al., 2016*; *Pereira et al., 2015*; *Ryan et al., 2014*; *Sammut et al., 2015*). This predicts that a number of additional unidentified changes may occur in the anatomy, connectivity and gene expression profiles of the male nervous system as the animal transitions from larva to adult. Given this, we asked if *daf-7* expression in males was correlated with the developmental stage of the animal. We observed that in populations of early larval animals carrying the *pdaf-7::gfp* transcriptional reporter, males and hermaphrodites were indistinguishable, with fluorescence observed only in the ASI neurons (*Figure 1—figure supplement 1D*). To determine the onset of *daf-7* expression in males, we imaged animals carrying the fluorescent reporter at multiple time points during development beginning in the L4 larval stage and extending into reproductive maturity. We found that *daf-7* expression remains off in the ASJ of males through the end of the last (L4) larval stage, but is switched on during young adulthood as the animals reach reproductive maturity (*Figure 1C*). In addition, we examined expression of *daf-7* in the ASI neuron pair at several matched timepoints during development and observed a significant drop in the expression of *daf-7* in the ASI neurons from L4 to young adulthood (*Figure 1—figure supplements 1E and F*, 52 hr and 64.5 hr timepoints). At subsequent timepoints during adulthood, however, expression levels in the ASI neurons remained constant as expression in the ASJ neurons increased (*Figure 1—figure supplements 1E*,3F 64.5 hr and 71.5 hr timepoints).

### Genetic sex of the nervous system modulates *daf-7* expression in adult males

We asked what the role of the genetic sex of the animal is in regulating *daf-7* expression in the ASJ neurons. The *C. elegans* sex determination pathway terminates on the master regulator GLI-family transcription factor, TRA-1 (*Zarkower and Hodgkin, 1992*; *Zarkower, 2006*). TRA-1 activity

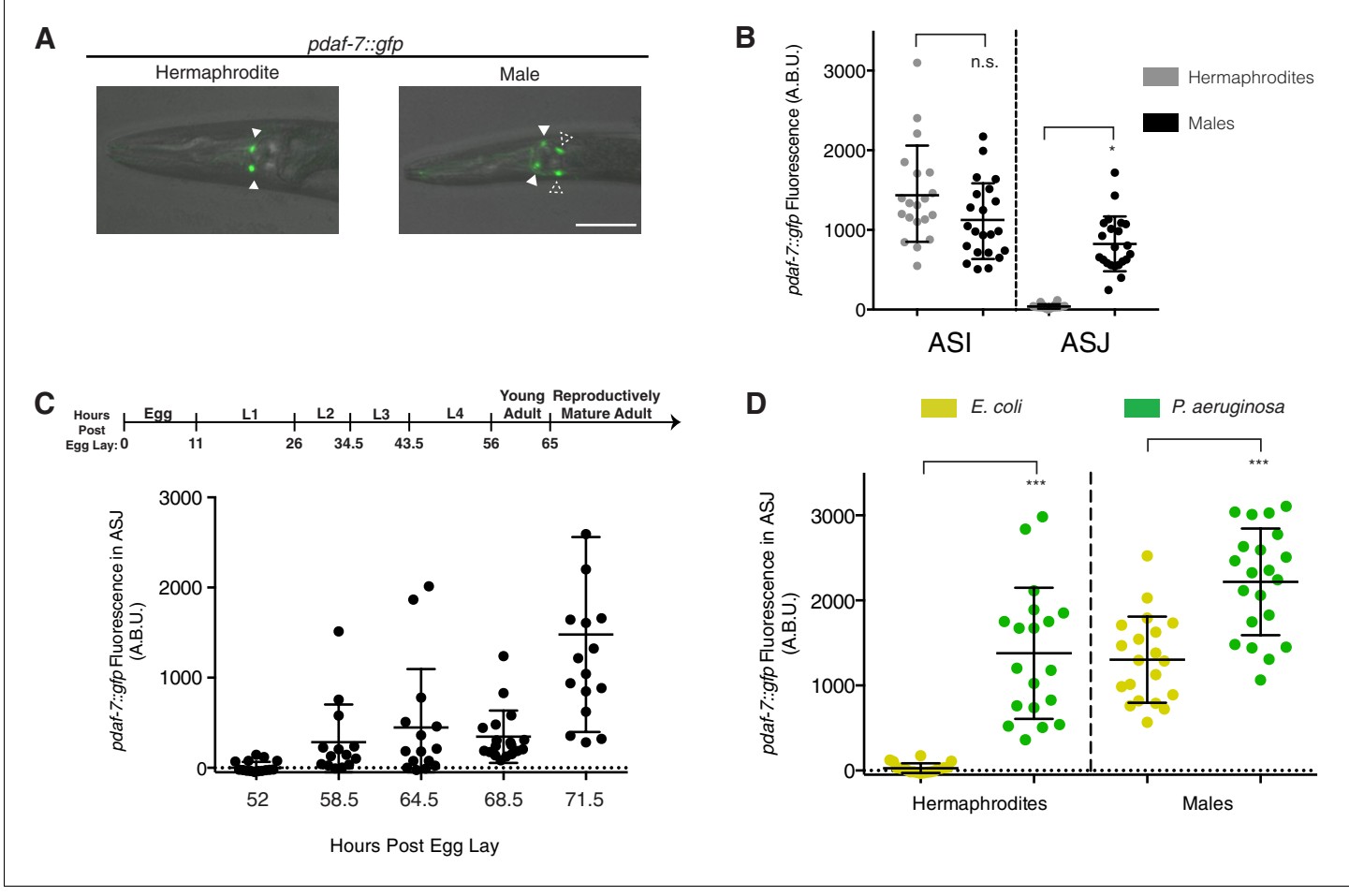

**Figure 1.** Sex-specific expression of *daf-7* in the ASJ neurons of adult males. (A) *pdaf-7::gfp* expression pattern in hermaphrodites (left) and males (right). Filled triangles indicate the ASI neurons; dashed triangles indicate the ASJ neurons. Scale bar indicates 50 μm. (B) Maximum fluorescence values of *pdaf-7::gfp* in the ASI (left) and ASJ (right) neurons of age-matched adult hermaphrodites (grey) and males (black). *p<0.05 as determined by an unpaired t-test with Welch's correction. Error bars indicate standard deviation (SD). n.s., not significant. (C) Maximum fluorescence values of *pdaf-7::gfp* in the ASJ neurons of males through larval development and early adulthood. The developmental timeline of *C. elegans* is shown to scale on top. Time values indicate hours after an egg is laid. Error bars indicate SD. (D) Maximum fluorescence values of *pdaf-7::gfp* in the ASJ neurons of both males and hermaphrodites on *E. coli* (yellow) or *P.aeruginosa* (green). ***p<0.001 as determined by unpaired t-tests with Welch's correction. Error bars indicate SD.

The following figure supplement is available for figure 1:

**Figure supplement 1.** *daf-7* expression in the ASJ neuron pair is specific to adult males.

functions to promote hermaphrodite cell fates and inhibit male development. We first looked at *tra-1(e1099)* mutants which are karyotypically XX, but develop as fertile pseudomales due to the lack of active TRA-1 (*Hodgkin and Brenner, 1977*). In these mutant XX males, *daf-7* expression could be seen in both the ASI and ASJ neurons, suggesting that *daf-7* is regulated by the genetic sex determination pathway either directly or indirectly (*Figure 2—figure supplement 1A*).

We next looked at the role of genetic sex specifically in the nervous system using transgenic sexual mosaics. TRA-1 is negatively regulated by the FEM proteins—FEM-1, FEM-2, and FEM-3—and it has been demonstrated by many groups that overexpression of FEM-3 cDNA is sufficient to lead to 'masculinization' in otherwise hermaphrodite animals (*Mehra et al., 1999*; *Oren-Suissa et al., 2016*; *Pereira et al., 2015*; *Sammut et al., 2015*; *White et al., 2007*). We expressed the FEM-3 cDNA under the control of the pan-neural promoter, *rab-3,* and found that *daf-7* expression was up-regulated in the ASJ neurons of adult hermaphrodites, suggesting that a genetically 'male' nervous

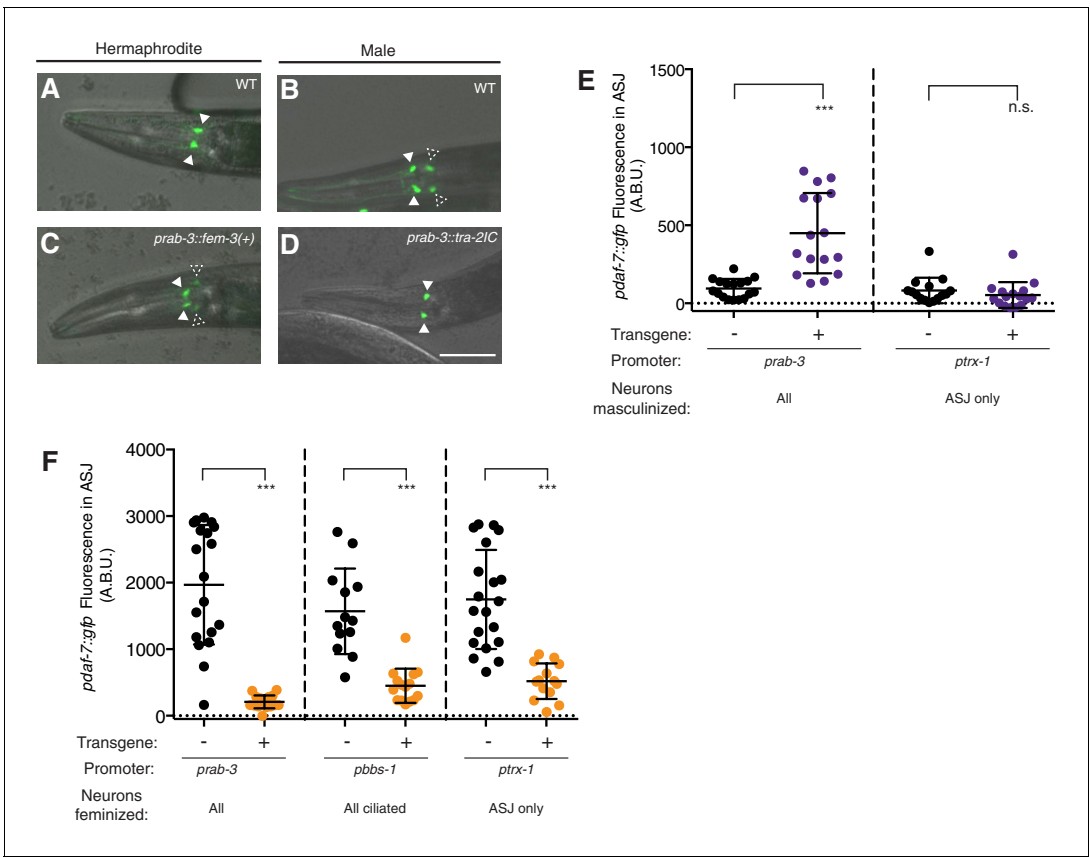

**Figure 2.** Genetic sex regulates male-specific expression of *daf-7* in the ASJ neuron pair. (A–D) Representative images of *pdaf-7::gfp* expression in: wild-type (WT) hermaphrodites (A) and males (B), nervous-system-masculinized hermaphrodites (C) and nervous-system-feminized males (D). Closed triangles indicate the ASI neurons; dashed triangles indicate the ASJ neurons. Scale bar indicates 50 μm. (E) Maximum fluorescence values of *pdaf-7::gfp* in the ASJ neurons of control (black) and partially masculinized hermaphrodites (purple). Masculinization was effected by driving expression of *fem-3* cDNA under pan-neural (left) and ASJ-specific (right) promoters. ***p<0.001 as determined by unpaired t-test with Welch's correction. n.s., not significant. Error bars represent SD. (F) Maximum fluorescence values of *pdaf-7::gfp* in ASJ of control (black) and partially feminized males (orange). Feminization was effected by driving expression of *tra-2*[IC] under pan-neural (left), ciliated neuron (middle), and ASJ-specific (right) promoters. ***p<0.001 as determined by unpaired t-test with Welch's correction. Error bars indicate SD.

The following figure supplement is available for figure 2:

**Figure supplement 1.** *daf-7* expression in the ASJ neuron pair is regulated by TRA-1 but not by male-specific neurons.

system is sufficient for *daf-7* expression in ASJ (*Figure 2A, C and E*). Although we observed an increase in *daf-7* expression in these genetically masculinized animals, the overall expression of *daf-7* in the ASJ of masculinized hermaphrodites was still notably lower than what we observe in wild-type males (e.g. *Figure 1B*). This could possibly be explained by the lack of male-specific neurons in these transgenic animals. While genetic masculinization is effective in sexualizing neurons of the sex-shared nervous system, this technique fails to reliably retain the male-specific CEM neurons and produces hermaphrodites which also lack other male-specific neurons found in the tail (*White et al., 2007*). To assess the importance of these male-specific neurons for *daf-7* expression in ASJ, we examined *pdaf-7::gfp* fluorescence in *ceh-30(n4289)*, *mab-3(e1240)*, and *pkd-2(sy606);lov-1(sy582)* mutant males. While *ceh-30(n4289)* males lack the male-specific CEM neurons in the head (*Schwartz and Horvitz, 2007*), *mab-3(e1240)* mutants display marked defects in male-tail formation including ray differentiation as well as in male specific circuitry in the head of the animal (*Yi et al., 2000*). Similarly, PKD-2 and LOV-1 constitute a TRP channel necessary for the function of the B-type ray neurons— another set of male-specific neurons in the tail of the animal (*Barr et al., 2001*; *Barr and Sternberg, 1999*; *Barrios et al., 2008*). Males of all mutant strains retained wild-type expression levels of *daf-7*

in ASJ, suggesting that the male-specific neural circuitry is not required for *daf-7* expression and is unlikely to underlie the lower *daf-7* expression we see in genetically masculinized hermaphrodites (*Figure 2—figure supplement 1B*).

Complementary to masculinization, genetic 'feminization' of XO male animals can be achieved via the overexpression of the intracellular domain of the TRA-2 protein, which negatively interacts with FEM-3 to prevent TRA-1 degradation (*Mehra et al., 1999*). Overexpression of TRA-2$^{IC}$ under the control of the *rab-3* promoter led to complete loss of *daf-7* expression in the ASJ of otherwise male animals (*Figure 2B, D and F*). These results suggest that a genetically male nervous system is both necessary and sufficient for the sex-specific expression pattern of *daf-7*.

We next asked whether the sex determination pathway acts cell-autonomously in ASJ to regulate *daf-7* expression. Using the *trx-1* ASJ-specific promoter, we masculinized the ASJ neuron pair alone and saw no *daf-7* expression in the resulting adult hermaphrodites (*Figure 2E*). Similarly, no *daf-7* expression could be observed when only the ciliated neurons (of which ASJ is a member) were masculinized (data not shown). These data suggest the requirement that an additional neuron (or neurons) or other tissues have a male cell identity for *daf-7* to be expressed in the ASJ neurons, although differences in the strength of the heterologous promoters used in the masculinization constructs may also account for these observations. In contrast, both ciliated neuron-specific and ASJ-specific feminization in males led to down-regulation of *daf-7* expression, confirming that a male cell identity in ASJ is necessary, though not sufficient, for proper *daf-7* expression (*Figure 2F*).

## DAF-7 is required for male-specific mate-searching behavior

We next sought to determine the functional significance of the sex-specific changes in *daf-7* expression that we observe in males. Mate-searching behavior is a behavioral program specific to adult male *C. elegans,* which features the prioritization of mating over feeding in these animals. While hermaphrodite animals prefer to remain in a patch of bacterial food indefinitely and exhibit limited exploratory behavior, males will wander away from a food source at a predictable and constant rate (*Lipton et al., 2004*). This can be examined in an assay in which the movement of the animal is tracked over time to determine the leaving probability of any given genotype (*Figure 3A*; *Lipton et al., 2004*). We observed that mutants in both the DAF-7/TGF-$\beta$ ligand and its Type I receptor, DAF-1, showed defects in the mate-searching assay (*Figure 3A and B*, and *Figure 3—figure supplement 1*). This behavioral defect in the *daf-7* mutant can be fully suppressed by mutation of the downstream antagonistic co-SMAD, DAF-3, suggesting that DAF-7 functions through its canonical TGF-$\beta$ signaling pathway to regulate mate-searching behavior (*Figure 3B*, and *Figure 3—figure supplement 1B*). Given the change in expression of *daf-7* in the ASJ neurons of males at the transition to adulthood when this behavior is first displayed, we used genetic ablations of the ASJ neuron pair to ask what their contribution to this behavior might be. We observed that the ASJ-ablated animals were defective in mate-searching behavior, establishing a role for the ASJ neuron pair in promoting this behavior (*Figure 3C* and *Figure 3—figure supplement 1A*). Importantly, the defects that we see in *daf-7* mutant and ASJ-ablated male behavior are not accounted for by global defects in locomotion. When mutant animals are tested on leaving assay plates on which the bacterial food has been fully spread, these males reach the scoring distance at equivalent rates to WT controls (*Figure 3—figure supplement 2*; *Barrios et al., 2012*). This suggests that the reduced leaving behavior we observe in these animals isspecific to the decision to leave the food source in search of a mate.

Since DAF-7 is a secreted ligand, we anticipated that overexpression of DAF-7 from either the ASI or ASJ neurons would be sufficient to rescue *daf-7*-associated mutant phenotypes. As expected, we saw that transgenic rescue from the endogenous promoter as well as from ASI- and ASJ-specific promoters were sufficient to restore mate-searching behavior in the *daf-7(ok3125)* mutant background (*Figure 3D* and *Figure 3—figure supplement 1C*). Taken together, these data imply that the dynamic expression of *daf-7* in the ASJ neurons of adult males regulates mate-searching behavior.

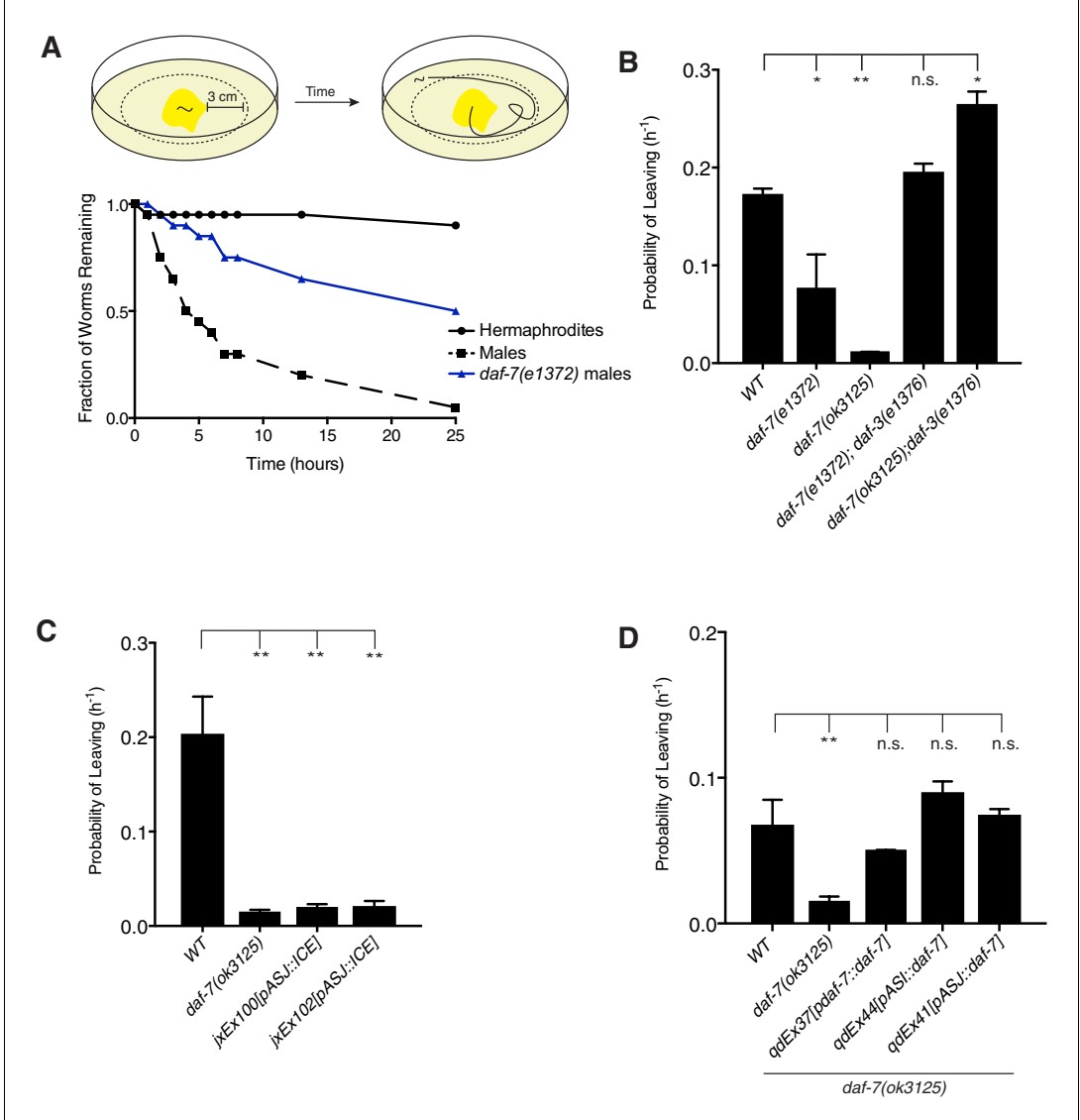

**Figure 3.** DAF-7/TGF-$\beta$ is required for male mate-searching behavior. (**A**) Schematic of mate-searching assay (top). Animals are placed individually into the center of a lawn of bacteria. The tracks of the animal are followed over time and scored for movement beyond 3 cm away from the food source. A representative data curve is shown on the bottom depicting hermaphrodites (solid black), males (dashed black), and *daf-7(e1372)* mutant males (blue). (**B**) Probability of leaving values for WT, *daf-7* mutant, and *daf-7;daf-3* double mutant males. Values plotted are the average + SEM for two independent experiments, n = 40 animals for all strains except *daf-7(ok3125)* where n = 29. *p<0.05, **p<0.01 as determined by ordinary one-way ANOVA followed by Dunnett's multiple comparisons test. n.s., not significant. (**C**) Probability of leaving values for two independent lines of male animals with genetic ablation of the ASJ neurons, compared with corresponding values for WT and *daf-7* mutant males. Values plotted are the average + SEM for two independent experiments for *daf-7* mutant animals and three independent experiments for WT control and ASJ ablation strains. n = 60 animals for all strains except *daf-7(ok3125)* where n = 29 animals. **p<0.01 as determined by ordinary one-way ANOVA followed by Dunnett's multiple comparisons test. One replicate was performed with the *daf-7;daf-3* double mutant strains in 1B and controls (WT and *daf-7*) are identical, so probability values for that experiment were used in the averages shown in both B and C of this figure. (**D**) Probability of leaving values for *daf-7(ok3125)* mutant animals carrying transgenes expressing *daf-7* cDNA specifically under the control of the indicated promoters. Values on graph are the average + SEM of two independent experiments with an n = 40 total animals for each strain. **p<0.01 as determined by ordinary one-way ANOVA followed by Dunnett's multiple comparisons test. n.s., not significant.

The following figure supplements are available for figure 3:

**Figure supplement 1.** *daf-7* regulates mate-searching behavior in males.

**Figure supplement 2.** *daf-7* mutant animals and strains carrying genetic ablation of the ASJ neurons exhibit no locomotion defect.

## Nutritional state dynamically regulates *daf-7* expression in the ASJ neurons of males

Mate-searching behavior has been demonstrated to be dependent on the feeding state of the animal—starved male animals will re-prioritize feeding over mate searching for a limited period of time until the animal has had sufficient opportunity to re-feed (*Figure 4C*; *Lipton et al., 2004*). We asked whether *daf-7* expression in the male ASJ neurons might be dependent on the nutritional state of the animal or the bacterial availability in its environment. We found that males subjected to starvation beginning early in the L4 larval stage failed to express *daf-7* in the ASJ neurons as adults (*Figure 4A*, conditions i and ii, and *Figure 4B*). This lack of *daf-7* in the ASJ neuron pair was apparent both by GFP visualization as well as by FISH, with *daf-7* mRNA entirely absent from the ASJ

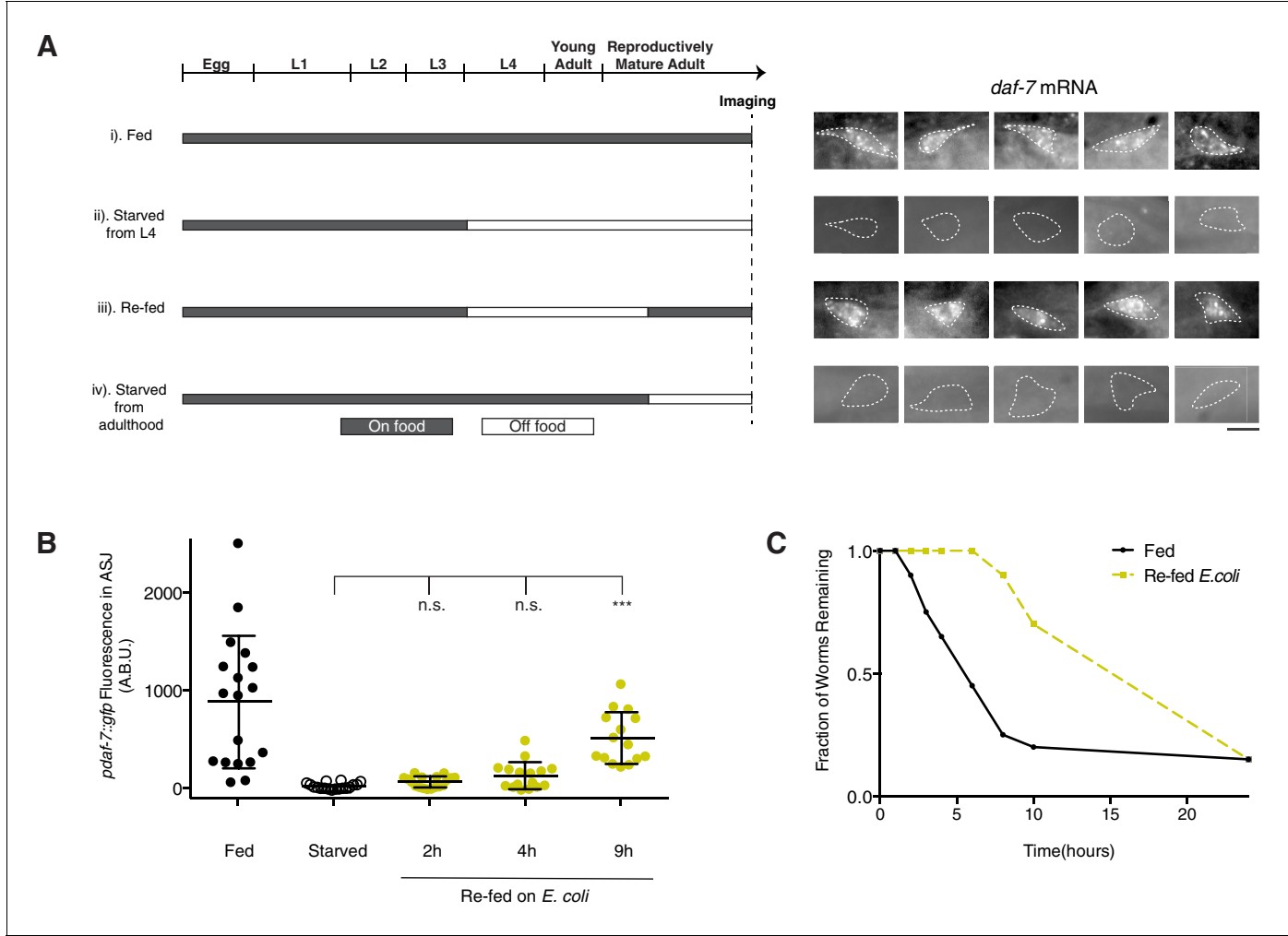

**Figure 4.** Nutritional state regulates *daf-7* expression in the ASJ neurons of males. (A) Schematic of experiment design indicating timing and duration of starvation periods (left). FISH images of endogenous *daf-7* mRNA in the ASJ neurons for each experimental condition (right). White dotted lines indicate outline of ASJ cell body as localized by *ptrx-1::gfp*. Scale bar represents 5 μm. (B) Maximum fluorescence values of *pdaf-7::gfp* in the ASJ neurons during re-feeding experiment. Re-fed animals were starved for a period of 24 hr before being reintroduced to *E. coli* for the indicated amount of time. ***p<0.001 as determined by ordinary one-way ANOVA followed by Dunnett's multiple comparisons test. Error bars indicate SD. n.s., not significant. (C) Representative curves of mate-searching data in fed and post-starvation WT males. Animals were starved for a period of 24 hr as in previous experiments before being assayed for mate-searching behavior.

The following figure supplement is available for figure 4:

**Figure supplement 1.** *daf-7* expression is down-regulated by starvation specifically in the ASJ neurons.

neurons (*Figure 4A and B*). We noticed that *daf-7* mRNA was still apparent in the ASI, OLQ, and ADE neurons of these starved animals, suggesting that the loss of *daf-7* expression in the ASJ neurons is specific and not representative of a global down-regulation of transcription in these animals (*Figure 4—figure supplement 1*).

We next asked if *daf-7* expression could be dynamically modulated by removal and re-introduction of food at various points during the life of the male. We first looked at *daf-7* expression in animals that had experienced starvation during the L4-to-adulthood period before being returned to an *E. coli* food source as adults. In these animals, *daf-7* expression could be detected both by the GFP reporter as well as by FISH (*Figure 4A*, condition iii, and 4B). We asked whether fully mature adult animals would turn off *daf-7* expression in response to starvation after having already induced expression of *daf-7* in the ASJ neuron pair. For these experiments, we utilized FISH of the endogenous *daf-7* mRNA because perdurance of the GFP reporter might complicate analysis. When males were allowed to develop into adulthood unperturbed and then subjected to a period of food deprivation, we observed a complete loss of *daf-7* mRNA in the ASJ neurons (*Figure 4A*, condition iv). Together this data demonstrates that *daf-7* expression in the ASJ neurons is subject to dynamic regulation by periods of starvation and re-feeding.

These data suggest two distinct possibilities for how bacterial food levels in the environment modulate *daf-7* transcriptional regulation in the ASJ neuron pair of males. First, *daf-7* transcription could be regulated by the internal nutritional state of the animal (starved or fed). Alternatively, *daf-7* transcription could be activated by the chemosensation (smell or taste) of bacterial food. To distinguish between these two possibilities, we measured the kinetics of *daf-7* expression in response to re-feeding in starved animals. We observed that *daf-7* expression in the ASJ neuron pair is not significantly increased compared to starved controls until 9 hr after the re-introduction of an *E. coli* food source (*Figure 4B*). Although some animals could be observed expressing *daf-7* weakly in the ASJ neuron pair at 4 hr of re-feeding, the majority of animals still exhibited no *pdaf-7::gfp* expression in the ASJ neuron pair until later time points. The extended period of time required to restore *daf-7* expression in the ASJ neuron pair suggests that internal nutritional state, not the chemosensation of bacterial food, exerts control over *daf-7* transcription in the ASJ neuron pair. Moreover, we observed that the onset of *daf-7* expression in the ASJ neurons of males following the re-introduction of food are consistent with the previously noted kinetics of resumption of mate-searching behavior after males have been starved (*Figure 4C*; *Lipton et al., 2004*).

## Differential regulation of *daf-7* expression in the ASJ neurons establishes a hierarchy of behaviors in male *C. elegans*

We recently showed that the microbial environment is an important regulator of *daf-7* expression in the ASJ neurons in hermaphrodite animals (*Figure 1D*; *Meisel et al., 2014*). Given that males induce expression of *daf-7* in the same ASJ neuron pair independently of the bacterial food source, we sought to determine if they would be capable of responding to exposure to *P. aeruginosa* in the same way as their hermaphrodite siblings—namely, through additional up-regulation of *daf-7* expression in the ASJ neurons. We observe that, when transferred to *P. aeruginosa*, males carrying the *pdaf-7::gfp* transcriptional reporter exhibited an increase in *daf-7* expression in the ASJ neurons compared to *E. coli*-fed controls (*Figure 1D*). The additive nature of DAF-7 levels in males suggests that the ASJ neuron pair of males retain the ability to respond to *P. aeruginosa* that we observe in hermaphrodites.

Our data suggest that *daf-7* expression in the ASJ neurons is subject to regulation by both sensory cues from the microbial environment as well as by internal state sensing mechanisms. Because *daf-7* expression in the ASJ neuron pair and mate-searching behavior in males are dependent on nutritional state during feeding on the non-pathogenic bacterial food, *E. coli*, we sought to determine if expression of *daf-7* in response to *P. aeruginosa* might also be dependent on the nutritional state in males. We tested this by challenging starved animals with *E. coli* or *P. aeruginosa* to see how *daf-7* expression would be affected and to observe any behavioral differences. Looking at animals carrying the *pdaf-7::gfp* reporter during post-starvation re-feeding, we observed a rapid up-regulation of *daf-7* expression in the ASJ neurons of starved males returned to *P. aeruginosa* as a food source (*Figure 5A and B*, and *Figure 5—figure supplement 1E*). This increase in *daf-7* expression in the ASJ neuron pair could be seen as early as 2 hr after first exposure to the pathogenic bacteria (*Figure 5—figure supplement 1E*). In contrast, starved males that were returned to the non-

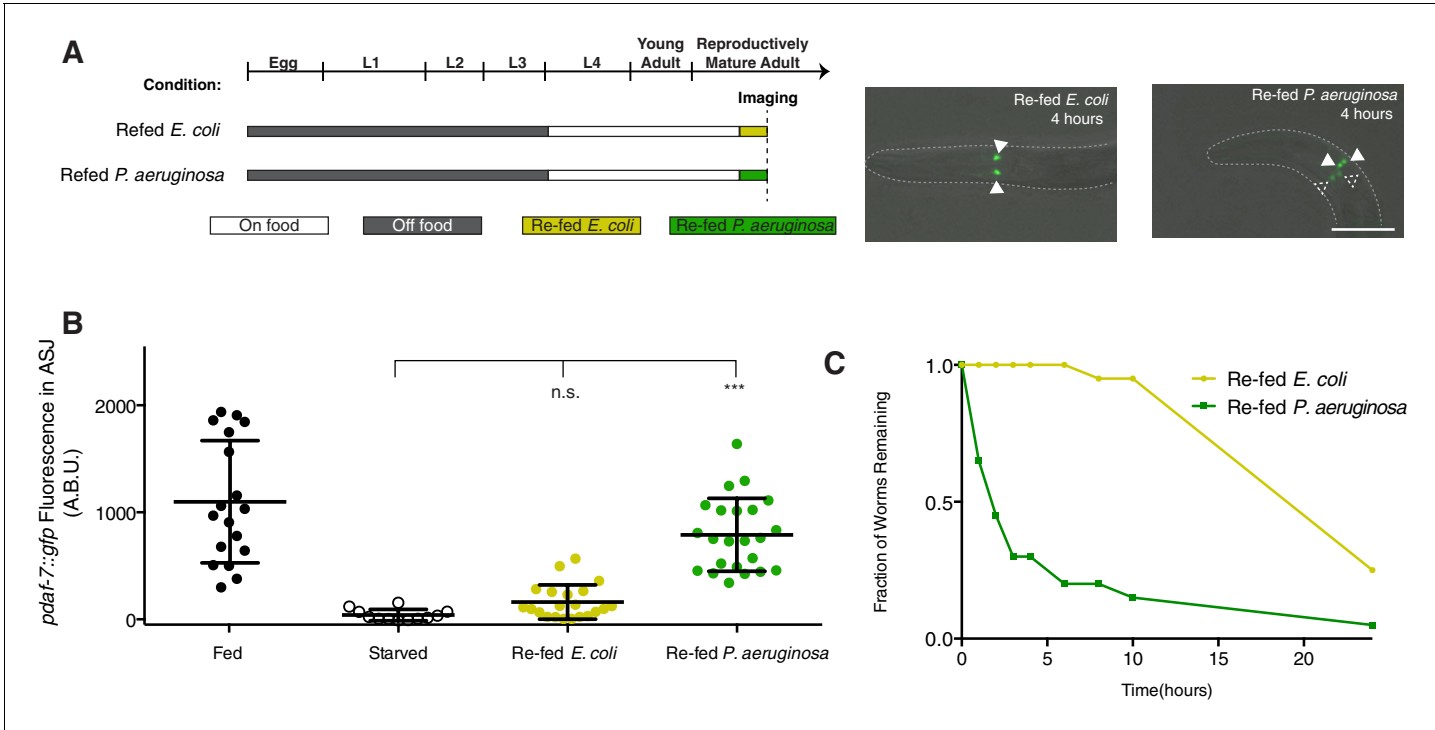

**Figure 5.** Prioritization of multiple inputs through the regulation of *daf-7* expression in the ASJ neurons of males. (**A**) Schematic of refeeding experimental design indicating timing and duration of each starving and feeding period (left). On right, representative images of *daf-7* expression following 4 hr of post-starvation refeeding on *E.coli* (left) or *P.aeruginosa* (right). Closed triangles indicate the ASI neurons; dashed triangles indicate the ASJ neurons. Scale bar represents 50 μm. (**B**) Maximum fluorescence values of *pdaf-7::gfp* in the ASJ neurons of fed and starved animals and animals that were starved for a period of 24 hr and then reintroduced either to the normal *E. coli* food source or to pathogenic *P. aeruginosa*. Images were taken for quantification after 4 hr of re-feeding. ***p<0.001 as determined by ordinary one-way ANOVA followed by Dunnett's multiple comparisons test. Error bars indicate SD. n.s., not significant. (**C**) Representative curve of food-leaving data in starved animals on *E. coli* (yellow) and *P. aeruginosa* (green). Animals were starved as in previous experiments before being run in the assay.

The following figure supplement is available for figure 5:

**Figure supplement 1.** *P. aeruginosa* influences exploratory behavior and *daf-7* expression in both males and hermaphrodites.

pathogenic *E. coli* showed little to no *daf-7* expression in the ASJ neurons until much later time points (*Figure 5A,B* and *4B*). The kinetics of the increase in fluorescence of the *pdaf-7::gfp* reporter in response to *P. aeruginosa* secondary metabolites, which lags behind the immediate response observed by FISH is consistent with what we observed previously for hermaphrodite animals (*Meisel et al., 2014*), and notably faster than the slower increase in *pdaf-7::gfp* reporter fluorescence in response to repletion of the non-pathogenic *E. coli* bacterial food.

We followed up by asking if this difference in expression after starvation is also reflected in behavioral differences of post-starvation animals experiencing different microbial environments. Normally, following a period of starvation, a male prefers feeding over leaving a lawn of bacteria in search of a mate (*Figure 4C*; *Lipton et al., 2004*), but we reasoned that these priorities might be reset if the only available food is pathogenic to the starved animal. Indeed, whereas starved males will remain on the *E. coli* lawn for several hours, starved males who are instead returned to a lawn of *P. aeruginosa* will readily leave the bacterial lawn at a rate similar to non-starved animals on the normal *E. coli* food source (*Figures 4C* and *5C*). This lawn-leaving behavior is not male-specific, as both fed and starved hermaphrodites will also readily leave a lawn of *P. aeruginosa* and starved hermaphrodites are also capable of responding to *P. aeruginosa* through an upregulation of *daf-7* in the ASJ neurons (*Figure 5—figure supplement 1A–1D*). In this regard, one might consider that lawn-leaving under these conditions is driven by the presence of pathogenic *P. aeruginosa*, and not mate-

searching. Our data suggest that *daf-7* expression in the ASJ neurons is one way through which animals are both able to interpret internal and external cues and prioritize their behaviors accordingly.

## Discussion

We have presented a set of experiments that suggest that behavioral prioritization and decision-making may be effected through the transcriptional control of a key neuroendocrine regulator, the TGF-*β* ligand, DAF-7, in a pair of chemosensory neurons, the ASJ neurons. Our data suggest that *daf-7* expression in the ASJ neuron pair is utilized to promote exploratory behaviors in different physiological and ecological contexts. In hermaphrodites, expression of *daf-7* is induced by the detection of secondary metabolites and functions to allow animals to discriminate between microbial species and promote avoidance of pathogenic bacteria (*Meisel et al., 2014*). In essence, pathogen avoidance represents the behavioral choice an animal makes when confronted with two competing needs—the need to eat and remain well fed and the need to avoid danger and infection. Mate searching presents a paralagous but sex-specific behavior, where the decision instead is between food and mating. Our data suggest that modulation of *daf-7* expression provides a striking example of how a neuroendocrine signaling pathway may be coopted to guide behavioral decisions in entirely different contexts.

Our study suggests that internal states and environmental cues that converge on the transcriptional control of DAF-7 in the ASJ neuron pair are processed in a hierarchical manner that, in turn, serves to modulate the exploratory mate-searching behavior in the male (*Figure 6*). The work presented here suggests that the transcriptional regulation of *daf-7* in the ASJ neurons functions as an ON/OFF switch, where each input holds the switch—and thus, *daf-7* expression—in one of two states. However, the various regulatory inputs into this switch exert differential control, with environmental cues being prioritized over all others. This differential control of the transcriptional switch, in turn allows for re-adjustments of behavioral priorities in the worm based on its current and past experience.

Interestingly, dynamic transcriptional regulation has previously been implicated in the regulation of sex-specific decision-making behavior in *C. elegans.* Expression of the chemoreceptor, ODR-10, in the sex-shared sensory AWA neurons has also been shown to be dependent on both internal and external cues to regulate food-related decision making in male *C. elegans* (*Ryan et al., 2014*). Given

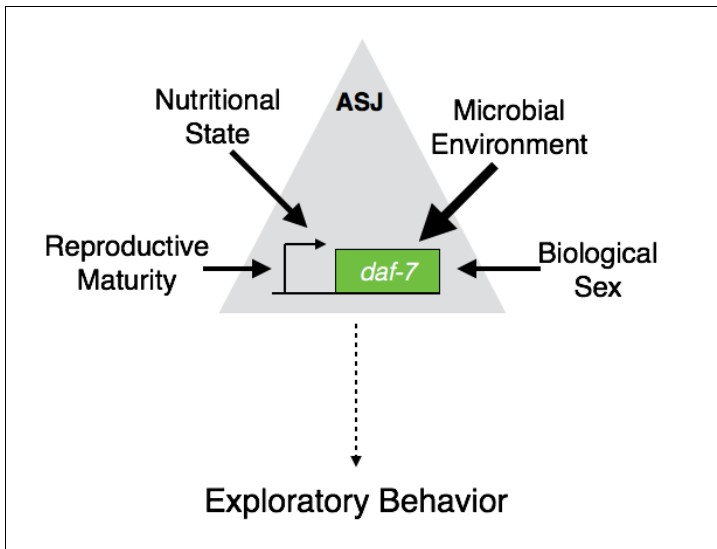

**Figure 6.** A hierarchy of inputs regulates *daf-7* expression in the ASJ neurons to modulate exploratory behaviors. We show that the information from many different sources both internal and external is integrated hierarchically in the regulation of *daf-7* expression in the ASJ neuron pair. This hierarchical regulation of *daf-7* in turn leads to a distinct prioritization of exploratory behaviors in the male worm.

the similarities in the regulation of these two genes by both sex and satiety, it is possible that DAF-7 and ODR-10 function in a single pathway to regulate decision-making behaviors in the male. Previous studies of *daf-7* have suggested that it is involved in regulating the expression of numerous chemoreceptors in many different neuron types (*Nolan et al., 2002*). While no interaction has been identified between *daf-7* and *odr-10* in the hermaphrodite nervous system, there has been no examination as yet of the effects of *daf-7* mutation on *odr-10* in the relevant cell types, sex, or under different food conditions, which may be an interesting line of future research.

DAF-7/TGF-$\beta$ has previously been found to be involved in the regulation of other satiety-regulated behaviors such as quiescence and foraging (*Gallagher et al., 2013*; *Milward et al., 2011*; *You et al., 2008*). *daf-7* mutants show marked defects in the ability to respond to fasting through an increase in quiescence, suggesting that DAF-7 may serve as a critical signal of food availability in the worm (*You et al., 2008*). Similarly, DAF-7 signaling has also been implicated in promoting food-leaving (or foraging) in the context of a rapidly depleting food source, further underscoring the importance of this pathway for communicating nutritional availability (*Milward et al., 2011*). While DAF-7 from the ASI neurons, and not the ASJ neurons, contributes to these other possibly related phenotypes, similar mechanisms may be involved in regulating *daf-7* expression and release in these three distinct food/satiety-dependent assay contexts. Taken together with our data, the important role of *daf-7* in these different behavioral paradigms suggests that the *daf-7* pathway may act as a more global food signal that is critical for allowing the animal to sense food and satiety and modulate its behaviors accordingly.

Finally, our results support a model where integration of internal and external sensory information can occur at the level of the sensory neuron itself to regulate behavioral prioritization. Our data on the role of the ASJ sensory neurons in male mate-searching behavior are consistent with emerging evidence from studies in several species to suggest that the regulatory control of sensory neurons contributes to information processing and decision making behavior (*Dey et al., 2015*; *Lebreton et al., 2015*; *Peckol et al., 2001*; *Ryan et al., 2014*).

## Materials and methods

### *C. elegans* strains

*C. elegans* were grown on *E. coli* OP50 as a food source as previously described (*Brenner, 1974*). Strains were grown at 20°C except for assays containing dauer-constitutive mutants, in which case, all strains were grown at 16°C until they passed the dauer developmental decision. Unless otherwise indicated, all strains carry the *him-5(e1490)* mutation to increase the number of spontaneous males available. For a complete list of strains used in this study, please see *Supplementary file 1*.

### Measurement of gene expression in the ASJ neurons

For quantification of *daf-7* expression in mutants and developing animals, animals were egg laid for 2 hr on OP50 to obtain synchronized populations of animals. The animals were then mounted and anesthetized in 50 mM sodium azide at the indicated hour of development or at 72 hr if not indicated. The animals were imaged at 40x using a Zeiss Axioimager Z1 microscope. 15–20 animals were imaged for each condition or strain. For quantification, maximum intensity values of GFP within the ASJ neurons was calculated using FIJI software (*Schindelin et al., 2012*); Fiji, RRID:SCR_002285).

For *P. aeruginosa* experiments, the *P. aeruginosa* PA14 strain was grown overnight at 37°C in 3 mL of LB as previously described and the next day 7 μL of culture was seeded onto SKA plates (*Tan et al., 1999*). Plates were incubated at 37°C overnight and then allowed to grow for an additional two days at room temperature before being used for experiments. Animals for *P.aeruginosa* experiments were synchronized by treatment with bleach and then allowed to hatch and arrest as L1 larvae in M9 media overnight. This provided synchronized populations free from any possible bacterial contamination. Animals were transferred to PA14 plates or OP50 controls at the L4 larval stage. Imaging and quantification were performed as described above at 16 hr after transfer to PA14 plates.

## Generation of transgenic animals

To generate the neural feminization and masculinization transgenes, the pGH8 plasmid containing the pan-neural *rab-3* promoter along with *mCherry::unc-54 3'UTR* was linearized by PCR. The *fem-3* and intracellular *tra-2* cDNAs were amplified from cDNA generated with an Ambion RETROScript kit. The *unc-54* 3'UTR was amplified from the pPD95.75 vector and the SL2 trans-splicing sequence leader (*Huang et al., 2001*; *Spieth et al., 1993*) was amplified from genomic DNA. The *fem-3* or *tra-2* cDNA along with the *unc-54* 3'UTR and SL2 sequence were cloned into the linearized pGH8 vector by Gibson assembly (*Gibson et al., 2009*). For cell specific manipulations, the ciliated neuron *bbs-1* promoter (1.9 kb) and ASJ specific *trx-1* promoter (1.1 kb; *Fierro-González et al., 2011*) were amplified from genomic DNA and inserted in the place of the *rab-3* promoter by Gibson assembly. All plasmids were sequenced to confirm identity and injected into *ksIs2;him-5(e1490)* animals at a concentration of 50 ng/μL, along with a plasmid carrying *ofm-1::gfp* (50 ng/μL). At least three independent transgenic lines were obtained and analyzed for each construct and one representative line is shown. For a complete list of primers used in this paper, please see *Supplementary file 2*.

## Mate-searching assays

Mate-searching behavior was assessed in animals through a food leaving assay as previously described (See diagram in *Figure 3A*; *Lipton et al., 2004*). Briefly, animals were placed individually onto a 1 cm lawn (~20 μL of culture) of *E. coli* OP50 in the center of a 10 cm plate. Movement of each animal was tracked at hourly time points. When an animal's tracks extended a distance of 3 cm away from the starting lawn, the animal was scored as a 'leaver'. It has been previously demonstrated that male leaving behavior can be modeled by the single exponential decay function $N(t)/N(0)=\exp(-\lambda t)$, where $N(t)$ refers to the number of non-leavers at a given time and the hazard value ($\lambda$) gives an estimate for the probability of leaving, which remains constant over the course of an assay (*Lipton et al., 2004*). Hazard values were calculated by fitting data using the R survival package with an exponential parametric survival model using maximum likelihood (R Project for Statistical Computing, RRID:SCR_001905). These values were pooled across replicates. At least two independent experiments were performed for each genotype (~40 animals per genotype). For leaving assays with *P. aeruginosa,* plates were supplemented with additional peptone and were seeded with 15 μL of bacterial culture. In addition, all plates for PA14 assays were incubated overnight at 37°C after being seeded with bacteria and then for an additional 2 days at room temperature before being used for the assay.

For locomotion assays, 10 cm mate-searching plates were seeded with ~400 μL of *E. coli* OP50 culture spread to the edge of the plate. Plates were allowed to dry and remained at room temperature for ~16 hr before being used in the assay. Males were placed individually on these plates and their movement tracked over time as in the standard mate-searching assay. The same scoring distance was used for locomotion assays as for the mate-searching assay.

## Fluorescence In Situ hybridization

FISH was performed as described previously (*Raj et al., 2008*). Animals were fixed in 4% formaldehyde for 45 min at 20°C. Following washing in PBS, animals were resuspended in 70% ethanol and incubated overnight at 4°C. Samples were then hybridized with probes to the endogenous *daf-7* mRNA as previously published (*Meisel et al., 2014*). Hybridization was carried out at 30°C overnight. Imaging was performed using a Nikon Eclipse Ti Inverted microscope with a Princeton Instruments PIXIS 1024 camera. All data were analyzed in FIJI. For quantifications of FISH data, fluorescence values were calculated within the ASJ (or ASI) neurons for the indicated conditions. All samples were hybridized using *daf*-7-Cy5 probes and imaged at an exposure time of 1500 ms. For all FISH experiments, animals carrying the *ofEx4[ptrx-1::gfp]* transgene were used in order to localize the ASJ neurons.

## Starvation experiments

For all starvation experiments, animals were synchronized by treatment with bleach and then were allowed to hatch and arrest as L1s in M9 media for 14 hr. After 14 hr, animals were dropped onto plates containing abundant OP50 *E. coli* and were allowed to develop for 34–36 hr at 20°C. At this time point, animals had already entered the L4 larval stage. All animals were washed from the plates

and subjected to an additional 4–5 washes with M9 media to remove any remaining bacteria. Animals were placed either back onto plates seeded with *E. coli* or were dropped onto plates lacking peptone to inhibit any bacterial growth.

For re-feeding experiments, animals were starved on peptone free plates for 24 hr at 20°C. After 24 hr of starvation, nearly all of the male animals had molted into adults as evident by tail and gonadal morphology. Only animals with adult morphology were used for experiments and imaging. They were subsequently collected and placed onto plates seeded with either *E. coli* or *P. aeruginosa* for the designated amount of time. For the FISH experiment shown in *Figure 3*, animals were re-fed on *E. coli* OP50 for 16 hr at 20°C before being fixed for imaging.

For the late starve experiments, animals dropped back onto *E. coli* after the washing step were allowed to grow for an additional 24 hr at 20°C before being collected and washed 4–5 times in M9 media. These animals were then placed onto peptone-free plates and starved for 16 hr at 20°C before being fixed for imaging.

## Statistical analyses

All statistical analysis was performed using the GraphPad Prism software (Graphpad Prism, RRID: SCR_002798). Statistical tests used are indicated in each figure legend.

## Acknowledgements

We thank HR Horvitz and the *Caenorhabditis* Genetics Center, which is funded by the NIH Office of Research Infrastructure Programs (P40 OD010440), for strains. We thank members of the Kim and Horvitz labs for helpful conversation and feedback on the manuscript and figures.

## Additional information

### Funding

| Funder | Grant reference number | Author |
| --- | --- | --- |
| National Institutes of Health | GM084477 | Dennis H Kim |

The funders had no role in study design, data collection and interpretation, or the decision to submit the work for publication.

### Author contributions

ZAH, Conceptualization, Formal analysis, Validation, Investigation, Visualization, Methodology, Writing—original draft, Writing—review and editing; DHK, Conceptualization, Supervision, Funding acquisition, Visualization, Methodology, Writing—original draft, Writing—review and editing

### Author ORCIDs

Zoë A Hilbert, http://orcid.org/0000-0002-3833-6912
Dennis H Kim, http://orcid.org/0000-0002-4109-5152

## Additional files

### Supplementary files

• Supplementary file 1. *C. elegans* strains used in this study A comprehensive list of the strains used in this study. With the exception of CB1490, all strains are previously unpublished and were constructed for this study.

• Supplementary file 2. Oligos used for transgenic strain generation. A comprehensive list of the oligos used for constructing the transgenic strains created for this study. All sequences are listed in the 5' to 3' direction.

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
