## [Decision Letter]

Thank you for submitting your article "Sexually Dimorphic Expression of *daf-7*/TGF-β in Sensory Neurons Regulates Behavior of *C. elegans*" for consideration by *eLife*. Your article has been reviewed by three peer reviewers, one of whom, Oliver Hobert, is a member of our Board of Reviewing Editors, and the evaluation has been overseen by a Senior Editor. The following individual involved in review of your submission has agreed to reveal her identity: Arantza Barrios (Reviewer #3).

The reviewers have discussed the reviews with one another and the Reviewing Editor has drafted this decision to help you prepare a revised submission.

All reviewers appreciated the overall importance of this work. Dynamic regulation of sensory neuron functions by external conditions and internal state is now appreciated as a major mechanism underlying behavioral plasticity. The ability to monitor the expression of single genes with defined functions in individual identified neurons in *C. elegans*, together with the ability to precisely manipulate external and internal conditions, has made this organism an excellent choice for exploring sensory neuron contributions to behavioral changes.

In this paper, the authors have followed up on their previous finding that the spatial expression pattern of the *daf-7* TGF-β ligand is dynamically regulated by perception of specific bacterial metabolites. You demonstrate that *daf-7* expression in the ASJ sensory neurons is also regulated by genetic sex, and that this sexually dimorphic expression contributes to male mate-searching behavior. Further regulation of expression by nutritional state allows reprioritization of feeding over mate-searching, a behavioral change previously reported by others. Overall, the experiments are nicely done, and the observation is novel. However, a number of concerns need to be addressed before this manuscript becomes acceptable for publication in *eLife*:

Major comments:

– One major concern is that this work is not well integrated with previously reported related work (Young-Jai You, Birgitte Olofsson) that implicated DAF-7 in regulating foraging behaviors in response to satiety state. Such integration would provide a more nuanced view that would make this work appropriate for publication in *eLife*. Several experiments are possible to integrate previous results on the roles of DAF-7 and ODR-10 in exploratory vs feeding behaviors. We suggest some specific ones here although other approaches may also be appropriate: (a) Compare time course of changes in *daf-7* vs *odr-10* expression upon starvation; (b) test if the changes in *daf-7* and *odr-10* are interdependent or independent of each other (i.e. analyze reporters in respective mutant backgrounds); (c) test if NPR-5 is involved in regulation of *daf-7* expression in ASJ in males as a function of satiety; (d) Do males raised on *P. aeruginosa* also exhibit increased food leaving?

– In Figure 5, starved males are shown to leave a lawn of *P. aeruginosa* at a higher rate than *E. coli*, and this is interpreted as mate-searching behavior. However, the leaving assay does not discriminate mate-searching behavior from what could be pathogen avoidance in this context. Either this assay needs to be performed on hermaphrodites to distinguish it as a male-specific behavior, or an additional assay (i.e. retention assay, as in Barrios et al. 2008) performed to distinguish between these two possibilities.

– Exploratory behaviors rely on the ability of worms to locomote properly. The authors need to show that ASJ-ablated and *daf-7* mutant males have no general defects in locomotion before they can conclude that these manipulations specifically affect the decision to explore within or away from the patch of food. One way by which this could be addressed is by measuring and comparing the rate at which wild-type and experimental males reach the scoring distance in a leaving-assay plate when the area is completely covered with food (as in Barrios et al. 2012, Nat. Neurosc. for example).

---

## [Author Response]

*Major comments:*

*– One major concern is that this work is not well integrated with previously reported related work (Young-Jai You, Birgitte Olofsson) that implicated DAF-7 in regulating foraging behaviors in response to satiety state. Such integration would provide a more nuanced view that would make this work appropriate for publication in eLife.*

We have included a more complete description of prior studies that have implicated a role for DAF-7 in behaviors modulated by satiety. In addition, we have investigated how genes previously implicated by the work of You and Olofsson in satiety-influenced behaviors affect the expression of DAF-7 in the ASJ neurons of males that promotes mate-searching behavior. Our studies in the development of the revised manuscript suggest that there is not a straightforward correlation that can be applied in the genetic analysis of the three distinct DAF-7-involved behavioral phenotypes of quiescence, foraging, and male mate-searching behavior.

*Several experiments are possible to integrate previous results on the roles of DAF-7 and ODR-10 in exploratory vs feeding behaviors. We suggest some specific ones here although other approaches may also be appropriate: (a) Compare time course of changes in daf-7 vs odr-10 expression upon starvation; (b) test if the changes in daf-7 and odr-10 are interdependent or independent of each other (i.e. analyze reporters in respective mutant backgrounds); (c) test if NPR-5 is involved in regulation of daf-7 expression in ASJ in males as a function of satiety; (d) Do males raised on P. aeruginosa also exhibit increased food leaving?*

We agree that suggested experiments linking *daf-7* and *odr-10* might be illuminating given prior work that implicates dynamic changes in *odr-10* expression in the sensory neurons of males. Because the reported changes in *odr-10* expression show a variable qualitative increase during starvation compared with the switch-like behavior of changes in *daf-7* expression in the ASJ neuron, we anticipate that DAF-7 may function upstream of the regulation of *odr-10* expression. We provide further discussion of possible connections between DAF-7 and the prior work of Ryan et al. on ODR-10 in the Discussion of our revised manuscript.

We tested whether *npr-5* mutants exhibited altered expression of *daf-7* in the ASJ neurons of males but did not observe any appreciable change under fed or starved conditions.

The reviewers suggest an experiment involving raising males on *P. aeruginosa*, but unfortunately, the pathogenicity of *P. aeruginosa* precludes such an analysis, as male and hermaphrodite animals raised in such a manner are sick and dying.

*– In Figure 5, starved males are shown to leave a lawn of P. aeruginosa at a higher rate than E. coli, and this is interpreted as mate-searching behavior. However, the leaving assay does not discriminate mate-searching behavior from what could be pathogen avoidance in this context. Either this assay needs to be performed on hermaphrodites to distinguish it as a male-specific behavior, or an additional assay (i.e. retention assay, as in Barrios et al. 2008) performed to distinguish between these two possibilities.*

We regret not providing a clear explanation of our rationale and interpretation of this experiment. Given our observation that starvation could abrogate the expression of *daf-7* expression in the ASJ neurons in males that would promote mate-searching under fed conditions, we sought to test whether starvation could also override our previously described induction of *daf-7* expression in the ASJ neurons in animals exposed to *P. aeruginosa*. We observe that males that have been starved maintain expression of *daf-7* in the ASJ neuron pair on *P. aeruginosa* and correspondingly exhibit food-leaving behavior—indeed, as the reviewer suggests, we interpret this as primarily driven by pathogen avoidance. We provide several additional panels (Figure 5—figure supplement 1), which show that this *P. aeruginosa* leaving behavior is executed by both males and hermaphrodites, fed and starved. Additionally, we did perform a retention assay, and saw that while hermaphrodites suppressed the leaving behavior of starved males on *E. coli*, their presence was not sufficient to suppress *P. aeruginosa* leaving behaviors. We have provided further clarification and discussion in the text to clarify the rationale and interpretation of this important experiment.

*– Exploratory behaviors rely on the ability of worms to locomote properly. The authors need to show that ASJ-ablated and daf-7 mutant males have no general defects in locomotion before they can conclude that these manipulations specifically affect the decision to explore within or away from the patch of food. One way by which this could be addressed is by measuring and comparing the rate at which wild-type and experimental males reach the scoring distance in a leaving-assay plate when the area is completely covered with food (as in Barrios et al. 2012, Nat. Neurosc. for example).*

We thank the reviewers for this suggestion. We performed leaving assays with the *daf-7* mutant and ASJ ablated strains in which the plate was covered with bacteria, and observed no difference in the ability of mutant animals to reach the scoring distance. These data are provided as Figure 3—figure supplement 2.